# Indoor Localisation for Detecting Medication Use in Parkinson's Disease

## Abstract

Parkinson's disease (PD) is a slowly progressive debilitating neurodegenerative disease which is prominently characterised by motor symptoms. Indoor localisation, including its in-home mobility features, could provide a digital biomarker that can be used to quantify how mobility changes as this disease progresses. To improve the effectiveness of current methods for indoor localisation, a transformer-based approach utilising multiple modalities, Received Signal Strength Indicator (RSSI) and accelerometer data from wearable devices, which provide complementary views of movement, is proposed. To properly evaluate our proposed method, we use a free-living dataset where the movements and mobility are greatly varied and unstructured as expected in real-world conditions. 12 pairs of people (one with PD, and the other a control participant) lived for five days in a smart home with various sensors. Our evaluation on such a dataset, which includes subjects with and without PD, demonstrates that our proposed network outperforms the current state-of-the-art in indoor localisation. We also show how the accurate room-level localisation predictions can be transformed into in-home mobility features (i.e. room-to-room transition duration) which can be used to effectively classify whether the PD participant is taking their medications or withholding them (increasing their symptoms).

## 1 Introduction

Parkinson's disease (PD) is a neurodegenerative disorder that affects around six million people worldwide. It is a chronic disease with four main symptoms: tremor, bradykinesia (slowness of movement), rigidity, and postural instability Jankovic (2008). Although PD is a slowly progressive disease, the symptoms can fluctuate hourly depending on medication intake timing, stress and other factors. These fluctuations make it challenging to capture the slow symptom progression in an individual (needed for example to measure how a new therapy changes disease progression) accurately over time using infrequent clinic or lab-based "snapshot" clinician-patient interactions. Also, to help a clinician understand the impact of PD on a patient, continuous monitoring is needed to give an accurate evaluation of how severe the symptoms and their fluctuations are for that individual. One method to determine whether sensor based monitoring can detect symptom fluctuations in PD is by evaluating the patients in an "OFF" medication state (i.e. when they have not taken their medication) since, when they withhold medications, patients tend to experience a worsening of symptoms. For example, when "OFF" medications, motor symptoms can become more severe which may hinder the subject's gait and movement around their own home. As a result, they may typically need more time to transition between rooms.

Indoor localisation can be used to measure transitions between rooms in a home. Knowing how slow or fast a person transitions between rooms may be able to predict whether a person is in an "OFF" medication state or not and, longer-term, interpret their symptoms' severity (e.g. de novo freezing of gait in a hallway could indicate symptom progression) Bachlin et al. (2009). Localisation can also add context in the measurement of other behaviours such as urinary function monitoring (e.g., how many times someone visits the bathroom overnight) which can impact the quality of life with PD He et al. (2016).

To perform indoor localisation in home environments, IoT-based platforms with multimodal sensors combined with machine learning can be used to provide an unobtrusive and continuous localisation.

Typically, a wearable inertial measurement unit (IMU) is used to produce a received signal strength indicator (RSSI) at points throughout a home, which can be used to create radio-map features for room-level localisation. To provide more accurate localisation, accelerometer data from the same wearable can also be used as it provides a means to distinguish different activities (e.g., walking vs standing). As some activities are tied to particular rooms (e.g. stirring a pan on the hob must be in a kitchen), accelerometer data may enrich RSSI in differentiating adjacent rooms, which RSSI alone may struggle with Jovan et al. (2022).

If accelerometer data are to provide extra features for separating adjacent rooms, greater consideration must be given to data generalisation across different PD patients. As PD is a heterogeneous disease, the severity of symptoms may vary from one patient to another Greenland et al. (2019). These severe symptoms, such as tremor, may affect the generalisation of accelerometer data, especially those worn on the patient's wrists, which is a common and well accepted placement location. Naively combining the accelerometer data with the RSSI may impair the performance of indoor localisation due to differing levels of tremor manifesting in the acceleration signal. In this work, we make two main contributions. (1) We describe the utilisation of RSSI enriched by the accelerometer data to perform room-level localisation. Similar to Jovan et al. (2022), our proposed network intelligently chooses accelerometer features which may improve the RSSI performance in performing indoor localisation. To properly evaluate our proposed method, we use a free-living dataset where the movements and mobility are greatly varied and unstructured as expected in real world conditions. Our evaluation on such a dataset, which includes subjects with and without PD, demonstrates that our proposed network outperforms the current state-of-the-art DCMN (Jovan et al. (2022)) in all categories. (2) We also show how the accurate room-level localisation predictions can be transformed into in-home mobility features (i.e. room-to-room transition duration) which can be used to effectively classify the "OFF" or "ON" medication state of a PD patient.

## 2    RELATED WORK

Early work in machine learning for monitoring a long-term neurodegenerative disease like PD started with a simple PD classification Fraiwan et al. (2016) or easy-to-distinguish symptoms identification Arora et al. (2015); Fisher et al. (2016) using one single sensor modality such as vision sensors for simplicity Li et al. (2018). As the research progressed, multiple sensors have been considered for performance improvement in identifying PD symptoms. For example, Heidarivincheh et al. (2021) utilised vision and accelerometer sensors available in a smart home to classify PD from non-PD. Masullo et al. (2020) match video sequences of silhouettes to accelerations from wearable sensors for a person re-indentification in a home environment. Notably, all of this research uses vision as their main data source; although powerful and rich in features, vision sensors typically raise privacy challenges in home settings and are typically absent in key rooms within the home for this reason.

RSSI data produced from wearable devices is one such mechanism with less privacy concerns. In indoor localisation, fingerprinting using RSSI is the typical technique used to estimate the location of wearable devices by exploiting this signal that can be measured in the environment. RSSI signals are not stable, they fluctuate randomly due to shadowing, fading and multi-path effects. However, many techniques have been proposed in recent years to tackle these fluctuations, and, indirectly, improve the localisation accuracy. Some of the work, Zhang et al. (2016), utilise deep neural network (DNN) to generate coarse positioning estimates from RSSI signals, which are then refined by a hidden Markov model (HMM) to produce a final estimate location. Other works, Ibrahim et al. (2018), try to utilise a time-series of RSSI data and exploit the temporal connections within each access point to estimate room-level position. A CNN is used to build localisation models to further leverage the temporal dependencies across time-series readings.

It has been suggested that we cannot rely on RSSI alone for indoor localisation in home environments for PD subjects due to shadowing rooms with tight separation Pandey et al. (2021). Jovan et al. (2022) et al. combine RSSI signals (for location estimate) and accelerometer data (for body movement estimate) to produce a more accurate location estimate by utilising a wider range of features to differentiate adjacent rooms. Their proposed network has shown an improvement in tracking a person with PD in a hallway that, in turn, can be used to track their transition time between rooms. Our work is inspired by Jovan et al. (2022) with two main improvements. While Jovan et al. capture

local temporal patterns of RSSI and accelerometer data via recurrent neural network, we capture the patterns through the use of causal convolutional neural networks (CNN). In addition, unlike Jovan et al. who stop at predicting room-to-room transition durations, we go a step further and use room-to-room transition behaviours, as input features to a simple binary classifier, to predict whether the PD participants is taking their medications or withholding them.

## 3 COHORT AND DATASET

**Dataset.** This dataset was collected using wristband wearable sensors, one on each wrist of all participants, containing tri-axial accelerometers and 10 Access Points (APs) placed through the residential home, each measuring the RSSI Kozlowski et al. (2018). The wearable devices wirelessly transmit data using the Bluetooth Low Energy (BLE) standard which can be received by the 10 APs. Each AP records the transmitted packets from the wearable sensor which contains the accelerometer readings sampled at 30Hz, with each AP recording RSSI values sampled at 5 Hz.

The dataset contains 12 pairs of participants living freely in a smart home for five days. Each pair consists of one person with PD and one person as the healthy control volunteer (HC). This pairing was chosen to enable PD vs HC comparison, for safety reasons for the person with PD, and also to increase the naturalistic social behaviour, particularly amongst the spousal pairs who already lived together. From the 24 participants, five females and seven males have PD. The average age of the participants is 58.5 and the average time since PD diagnosis for the person with PD is 11.3 years. To measure the accuracy ML models, wall-mounted cameras are installed in the ground floor of the house as ground truth which capture red-green-blue (RGB) and depth data during the day for 2-3 hours daily. These cover the kitchen, hallway, dining room, living room, stairs, and porch. The duration of data recorded by the cameras for PD and HC is 72.84 and 75.31 hours, respectively, which provides a relatively balanced label set for our room-level classification. Finally, to evaluate the "OFF" medication state, participants with PD were asked to withhold their dopaminergic medications so that they were in the practically-defined "OFF" medications state for a period of several hours within five days.

**Data pre-processing.** The two wearable sensors worn by each participant are grouped together based on their types, i.e., twenty RSSI values corresponding to 10 APs for each wearable sensor, and six spatial directions corresponding to three spatial directions (x, y, z) for each wearable, at each time. The accelerometer data is resampled to 5Hz to synchronise the data with RSSI value. With a 5-second time window, RSSI has an input of size (25 x 20) and accelerometer data has an input of size (25 x 6). Imputation for missing values, specifically for RSSI data, is applied by replacing the missing values with a value that is not possible normally (i.e., -120dB). Both RSSI and accelerometer data were ignored when participants were out of the house. Finally, all timeseries measurements by the types are normalized to be within the range of zero and one before they are processed by a neural network model.

Our main focus is for our neural network model to produce in-home mobility features particularly for persons with PD. We hypothesize that during their "OFF" medication state, the deterioration in mobility of a person with PD is exhibited by how they transition between rooms. These features include 'Room-to-room Transition', and the 'Number of Transitions' between two rooms. With the layout of the house where participants stayed, the hallway is used as a hub connecting any other room on the ground floor, and 'Room-to-room Transition' shows the transition duration (in seconds) between two rooms connected by the hallway. The transition between (1) kitchen and living room, (2) kitchen and dining room, and (3) dining room and living room are chosen as the features due to their commonalty across all participants. For these features, we limit the transition time duration (i.e. the time spent in the hallway) to 60 seconds.

These features are produced by each model through the use of all available data from 12 PD participants including unannotated data from 6 a.m. to 10 p.m. daily which are aggregated into 4 hour window. From this, each PD participant will have 20 samples (four samples for each day), each of which contains six features (three for room-to-room transition duration, and three for the number of room-to-room transitions), and only one of which represents "OFF" medication state. These fea-

tures are then fed to a simple binary classifier[1] determining whether a person with PD is on their medication or not.

# 4 ROOM-LEVEL LOCALISATION WITH MULTI-SENSORY INPUTS

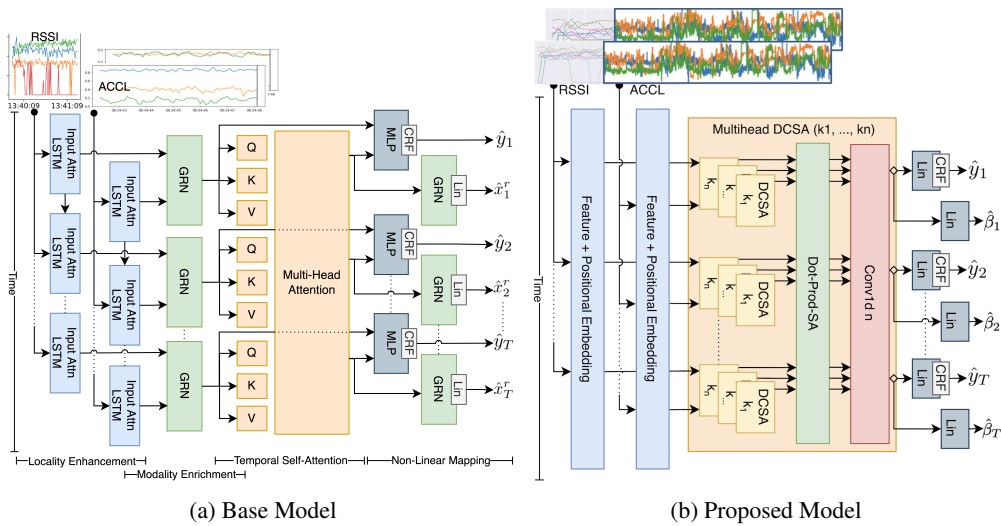

(a) Base Model      (b) Proposed Model

Figure 1: Overview of the base model of Jovan et al. (2022) against our proposed model.

**The Base Model.** As our base model, we take an approach by Jovan et al. (2022) which proposes a technique to utilise dual modalities (i.e. utilising two different sensor types) for indoor localisation in home environments. The approach can accurately predict room-level localisation by reducing any room-to-room transition inaccuracies by integrating an extra modality (i.e. accelerometer data) and its temporal aspects into the main modality (i.e. RSSI data). This approach, which is called Dual Context Modality Network (DCMN), is chosen due to its similarity in inputs (i.e. two types of sensors) and outputs (i.e. room-level localisation). It was shown to attain SOTA localisation performance on their dataset.

Suppose we have a collection of RSSI signals $\mathbf{x}^r = \left[\mathbf{x}_1^r, \ldots, \mathbf{x}_T^r\right] \in \mathbb{R}^{T \times r}$, and accelerometer data $\mathbf{x}^a = \left[\mathbf{x}_1^a, \ldots, \mathbf{x}_T^a\right] \in \mathbb{R}^{T \times a}$ within $T$ time unit, where $\mathbf{x}_t^r = \left[x_t^1, \ldots, x_t^r\right]$ represents RSSI signals from $r$ access points, and $\mathbf{x}_t^a = \left[x_t^1, \ldots, x_t^a\right]$ represents accelerometer data from $a$ spatial directions at time $t$ with $t \leqslant T$. During training, DCMN gets $\mathbf{x}^r$ and $\mathbf{x}^a$ as inputs, and produces reconstructed RSSI signals $\hat{\mathbf{x}}^r = \left[\hat{\mathbf{x}}_1^r, \ldots, \hat{\mathbf{x}}_T^r\right] \in \mathbb{R}^{T \times r}$ and embeddings of the predicted room locations (outputs before they go through Conditional Random Field layer) $\hat{\mathbf{e}}^\mathbf{y} = \left[\hat{\mathbf{e}}_1, \ldots, \hat{\mathbf{e}}_T\right] \in \mathbb{R}^{T \times d}$ with $d$ as the embedding dimension, i.e.:

$$(\hat{\mathbf{e}}^\mathbf{y}, \hat{\mathbf{x}}^r) = DCMN(\mathbf{x}^r, \mathbf{x}^a) \tag{1}$$

Jovan et al. (2022) suggested $\hat{\mathbf{e}}^\mathbf{y}$ is trained to generate the likelihood estimate of room predictions, while $\hat{\mathbf{x}}^r$ is used in an auto-encoding fashion to enhance the representation power of RSSI data. The final loss function is a combination of both likelihood and reconstruction losses:

$$\mathcal{L}(\hat{\mathbf{e}}^\mathbf{y}, \mathbf{y}, \hat{\mathbf{x}}^r, \mathbf{x}^r) = \mathcal{L}_{NLL}(\hat{\mathbf{e}}^\mathbf{y}, \mathbf{y}) + \sum_{i=1}^{T} \mathcal{L}_R(\hat{x}_i^r, x_i^r) \tag{2}$$

with $\mathcal{L}_{NLL}(.)$ represents the negative log-likelihood and $\mathcal{L}_R(.)$ denotes the reconstruction loss.

**The Issues.** Although DCMN Jovan et al. (2022) has a good attempt at utilising multiple sensor data to perform indoor localisation, there are several issues with the technique that limit its performance:

1. **The Lack of Local Context Matching with LSTM.** Patterns in time series may evolve and can be repeated with time due to various events (e.g. RSSI signals that fluctuate due

---

[1]The Random Forest is chosen as a binary classifier; It is not the one used for indoor localisation.

to shadowing), so whether an observed point is an anomaly (e.g. a sudden drop of one of the RSSI signals) or part of the patterns is highly dependent on its surrounding context. However, in LSTM layers, the local context is summarised based on the previous summarised context and the current input. Two similar patterns separated by a long period of time might have different context if they are processed by the LSTM layers Bahdanau et al. (2014). Furthermore, when two summarised contexts at different time points arrive at the MultiHead Self-Attention (MHSA) layer, even though they might be of similar patterns, the query-key matching of MHSA will not be able to capture the similarity due to their point-wise computation.

2. **The Mix-Matched of Local Context Between Two Sensors.** Figure 1a shows that the inputs to the modality enrichment layer are the summarised RSSI context and the summarised accelerometer context at each time step $t$. Since there is no way to fix the length of local context that the LSTM can squeeze, the summarised context contain variable lengths of previous information. This varies not only among different time step $t$ but also between RSSI signals and the accelerometer data at the same time step $t$.

3. **Unnecessary Reconstruction Loss.** In Equation 2, the reconstruction loss acts as a regularisation and also to capture the data distribution of the RSSI signals. However, to capture in-home mobility which heavily relies on correctly predicting hallway, reconstruction loss might work against predicting hallway due to relatively infrequent labels for the hallway compared to other room locations. We would, therefore, like to remove the use of reconstruction loss and add a different loss that improves the accuracy of predicting particular room locations with less labels in a home.

## 4.1 Locality Enhancement with Dual Convolutional Self-Attention

As it is time series data, the importance of an RSSI or accelerometer value at each point in time can be identified in relation to its surrounding values - such as cyclical patterns, trends, or fluctuations. Utilising historical context, that can capture local patterns on top of point-wise values, can thus lead to performance improvements in attention-based architectures. We consider to replace the LSTM on Figure 1a with a combination of *causal convolution* layers and a self-attention layers as Dual Convolutional Self-Attention (DCSA). The causal convolution layer was firstly introduced by Li et al. (2019) with kernel size $k$ and stride 1 to extract local patterns using the same filter across all time. Note that causal convolutions ensure that the current position never has access to future information. The DCSA takes in a primary input $\hat{\mathbf{x}}_1 \in \mathbb{R}^{T \times d}$ and a secondary input $\hat{\mathbf{x}}_2 \in \mathbb{R}^{T \times d}$ and yields:

$$DCSA_k(\hat{\mathbf{x}}_1, \hat{\mathbf{x}}_2) = GRN(Norm(\Phi(\hat{\mathbf{x}}_1) + \hat{\mathbf{x}}_1), Norm(\Phi(\hat{\mathbf{x}}_2) + \hat{\mathbf{x}}_2)) \tag{3}$$

with

$$\Phi(\hat{\mathbf{x}}) = SA\Big(\Phi_k(\hat{\mathbf{x}})\,\mathbf{W_Q}, \Phi_k(\hat{\mathbf{x}})\,\mathbf{W_K}, \hat{\mathbf{x}}\,\mathbf{W_V}\Big) \tag{4}$$

where $GRN(.)$ is Gated Residual Network (GRN), introduced by Lim et al. (2019), to integrate dual inputs into one integrated embedding, $Norm(.)$ is a standard layer normalisation, $SA(.)$ is a scaled dot-product self-attention introduced in Vaswani et al. (2017), $\Phi_k(.)$ is a 1D-convolutional layer with a kernel size $\{1, k\}$ and a stride 1, $\mathbf{W_K} \in \mathbb{R}^{d \times d}$, $\mathbf{W_Q} \in \mathbb{R}^{d \times d}$, $W_\mathbf{V} \in \mathbb{R}^{d \times d}$ are weights for keys, queries and values of the self-attention layer, and $d$ is the embedding dimension. Note that all weights for GRN are shared across each time step $t$.

## 4.2 Multihead Dual Convolutional Self-Attention

Similar to the base model, our approach employs a self-attention mechanism introduced by Vaswani et al. (2017) to capture global dependencies across time steps. It is embedded as part of the DCSA architecture. Inspired by Vaswani et al. (2017) in utilising multi self-attention, we utilise our DCSA with various kernel length with the same aim: allowing asymmetric long-term learning. The multihead DCSA, shown as part in Figure 1b, takes in two inputs $\hat{\mathbf{x}}_1, \hat{\mathbf{x}}_2 \in \mathbb{R}^{T \times d}$ and yields:

$$MDCSA_{k_1,...,k_n}(\hat{\mathbf{x}}_1, \hat{\mathbf{x}}_2) = \Phi_n(\Phi_{k_1,...,k_n}(\hat{\mathbf{x}}_1, \hat{\mathbf{x}}_2)) \tag{5}$$

with

$$\Phi_{k_1,...,k_n}(\hat{\mathbf{x}}_1, \hat{\mathbf{x}}_2) = SA(\Xi_{k_1,...,k_n}(\hat{\mathbf{x}}_1, \hat{\mathbf{x}}_2)\,\mathbf{W_Q}, \Xi_{k_1,...,k_n}(\hat{\mathbf{x}}_1, \hat{\mathbf{x}}_2)\,\mathbf{W_K}, \Xi_{k_1,...,k_n}(\hat{\mathbf{x}}_1, \hat{\mathbf{x}}_2)\,\mathbf{W_V})$$
$$\tag{6}$$

$$\Xi_{k_1,\ldots,k_n}(\hat{\mathbf{x}}_1, \hat{\mathbf{x}}_2) = [DCSA_{k_1}(\hat{\mathbf{x}}_1, \hat{\mathbf{x}}_2), \ldots, DCSA_{k_n}(\hat{\mathbf{x}}_1, \hat{\mathbf{x}}_2)] \tag{7}$$

where $\Phi_n(.)$ is a 1D-convolutional layer with a kernel size $\{1, n\}$ and a stride $n$, $\mathbf{W_K} \in \mathbb{R}^{d \times d}$, $\mathbf{W_Q} \in \mathbb{R}^{d \times d}$, $W_\mathbf{V} \in \mathbb{R}^{d \times d}$ are weights for keys, queries and values of the self-attention layer, and $\Xi_{k_1,\ldots,k_n}(.)$ concatenates the output of each $DCSA_k(.)$ in temporal order. For regularisation, a normalisation layer followed by a dropout layer is added after Equation 5.

**Feature Transformation.** Without LSTM layers acting as an encoder, a linear layer with a positional encoding is added to transform both RSSI and accelerometer data into their respective embeddings. The positional embeddings of RSSI $\mathbf{h}^r = [\mathbf{h}_1^r, \ldots, \mathbf{h}_T^r]$ and accelerometer $\mathbf{h}^a = [\mathbf{h}_1^a, \ldots, \mathbf{h}_T^a]$ are then fed to an MDCSA network with various kernel size $[k_1, \ldots, k_n]$:

$$\hat{\mathbf{h}} = MDCSA_{k_1,\ldots,k_n}(\mathbf{h}^r, \mathbf{h}^a) \tag{8}$$

### 4.3 Final Layer and Loss Calculation

We apply two different layers to produce two different outputs during training. The room-level predictions are produced via a single conditional random field (CRF) layer in combination with a linear layer applied to the output of Equation 8 to produce the final predictions as

$$\hat{y}_t = CRF\big(\mathbf{W}_p \hat{\mathbf{h}}_t + \mathbf{b}_p\big) \tag{9}$$

where $\mathbf{W}_p \in \mathbb{R}^{d \times m}$, and $\mathbf{b}_p \in \mathbb{R}^m$ are weight and bias to learn, $m$ is the number of room locations, and $\hat{\mathbf{h}} = [\hat{\mathbf{h}}_1, \ldots, \hat{\mathbf{h}}_T] \in \mathbb{R}^{T \times d}$ is the refined embedding produced by Equation 8.

For the second layer, instead of trying to reconstruct the RSSI and comparing it with the actual RSSI signals, we choose a particular room as a reference and perform a binary classification at each time step $t$. The binary classification is produced via a linear layer applied to the refined embedding $\hat{\mathbf{h}}$ as

$$\hat{\beta}_T = \mathbf{W}_\beta \hat{\mathbf{h}}_t + \mathbf{b}_\beta \tag{10}$$

where $\mathbf{W}_\beta \in \mathbb{R}^{d \times 1}$, and $\mathbf{b}_\beta \in \mathbb{R}$ are weight and bias to learn, and $\hat{\beta} = [\hat{\beta}_1, \ldots, \hat{\beta}_T] \in \mathbb{R}^T$ is the target probabilities for the referenced room within time window $T$. The binary classification loss can be formulated through the use of binary cross entropy loss function described as:

$$\mathcal{L}_{BCE}(\hat{\beta}, \beta) = -\frac{1}{T} \sum_{i=0}^{T} \beta_i \, log(\hat{\beta}_i) + (1 - \beta_i) \, log(1 - \hat{\beta}_i) \tag{11}$$

where $\beta = [\beta_1, \ldots, \beta_T] \in \mathbb{R}^T$ is the binary value whether at time $t$ the room is the referenced room or not. The final loss function is the Equation 2 with the $\mathcal{L}_R(.)$ being replaced by Equation 11.

## 5 Experiments and Results

We compare our proposed network with the base model of Jovan et al. (2022) as a SOTA representation for indoor localisation with dual modalities, Random Forest (RF) as a most baseline technique which has been shown to work well for indoor localisation, and TENER Yan et al. (2019) which is a modified transformer encoder in combination with a CRF layer. TENER represents a model with capability to capture global dependency and enforce dependencies across temporal aspects; hence, it can be considered as our algorithm without the locality enhancement – it can be regarded as ablations. As a completion for the ablation study, we also include $MDCSA_1$ network when our network only uses MDCSA with a single DCSA kernel of size $1^2$, and "Ours RSSI" network when our network only uses RSSI, without ACCL, as its input features. For RF, all time series features of RSSI and accelerometer are flattened and merged into one long feature vector for room-level localisation. For TENER, at each time step $t$, RSSI $\mathbf{x}_t^r$ and accelerometer $\mathbf{x}_t^a$ features are combined via a linear layer before they are processed by the networks. A grid search on the parameters of each network is performed to find the best parameter for each model. The parameters to tune are: the embedding dimension $d$ in $\{128, 256\}$, the number of epochs in $\{200, 300\}$, and the learning rate in $\{0.01, 0.0001\}$. The dropout rate is set to 0.15, and the RAdam optimizer Liu et al. (2019) in combination with Look-Ahead algorithm Zhang et al. (2019) is used for the training with early stopping using the validation performance. For RF, we perform a cross-validated parameter search for the number of trees ($\{200, 250\}$) and the minimum number of samples in a leaf node ($\{1, 5\}$). The Gini impurity is used to measure splits.

---

[2]Our regular network uses $MDCSA_{1,4,7}$ with 3 kernels of size 1, 4, and 7

Table 1: Room-level and medication state accuracy of all models. Standard deviation is shown under (.), the best performer is bold, while the second best is italicized.

| Training | Model | Room-Level Localisation | | Medication State | |
| | | Precision | F1-Score | F1-Score | AUROC |
|---|---|---|---|---|---|
| **ALL-HC** | RF | **95.00** | **95.20** | 56.67 (17.32) | **84.55 (12.06)** |
| | TENER | 94.60 | 94.80 | 47.08 (16.35) | 67.74 (10.82) |
| | MDCSA$_1$ | 94.80 | 95.00 | *62.11 (18.07)* | 80.20 (14.63) |
| | Ours RSSI | 94.70 | 94.90 | 51.14 (11.95) | 68.33 (18.49) |
| | DCMN | 94.80 | 95.00 | 47.25 (5.50) | 75.63 (4.49) |
| | Ours | *94.90* | *95.10* | **64.13 (6.05)** | *80.95 (10.71)* |
| | **Demographic Features** | | | 49.74 (15.60) | 65.66 (18.54) |
| **LOO-HC** | RF | 89.67 (1.85) | 88.95 (2.61) | 54.74 (11.46) | 69.24 (17.77) |
| | TENER | 90.35 (1.87) | 89.75 (2.24) | 51.76 (14.37) | 70.80 (9.78) |
| | MDCSA$_1$ | *90.67 (1.80)* | *89.99 (2.73)* | **64.70 (9.37)** | *77.26 (9.09)* |
| | Ours RSSI | 90.26 (2.43) | 89.48 (3.47) | *58.84 (23.08)* | 76.10 (10.84) |
| | DCMN | 90.52 (2.17) | 89.71 (2.83) | 49.56 (17.26) | 73.26 (10.65) |
| | Ours | **91.39 (2.13)** | **91.06 (2.62)** | 55.50 (15.78) | **83.98 (13.45)** |
| | **Demographic Features** | | | 51.79 (15.40) | 68.33 (18.43) |
| **LOO-PD** | RF | 86.89 (7.14) | 84.71 (7.33) | 43.28 (14.02) | 62.63 (20.63) |
| | TENER | 86.91 (6.76) | 86.18 (6.01) | 36.04 (9.99) | 60.03 (10.52) |
| | MDCSA$_1$ | *88.02 (6.98)* | *87.60 (5.94)* | **51.89 (15.32)** | *71.07 (10.64)* |
| | Ours RSSI | 87.61 (6.64) | 87.21 (5.44) | 45.71 (17.85) | 67.76 (10.73) |
| | DCMN | 87.36 (6.30) | 86.44 (6.63) | 44.02 (16.89) | 69.70 (12.04) |
| | Ours | **88.04 (6.94)** | **87.82 (6.01)** | *49.99 (13.18)* | **81.08 (8.46)** |
| | **Demographic Features** | | | 43.89 (14.43) | 60.95 (25.16) |
| **4m-HC** | RF | 74.27 (8.99) | 69.87 (7.21) | *50.47 (12.63)* | 59.55 (12.38) |
| | TENER | 69.86 (18.68) | 60.71 (24.94) | N/A | N/A |
| | MDCSA$_1$ | *82.65 (7.06)* | *78.21 (8.03)* | 48.10 (19.71) | *71.26 (12.43)* |
| | Ours RSSI | 81.69 (6.85) | 77.12 (8.46) | 49.95 (17.35) | 69.71 (11.55) |
| | DCMN | 78.79 (3.95) | 71.44 (9.82) | 43.89 (11.60) | 64.67 (12.88) |
| | Ours | **83.32 (6.65)** | **80.24 (6.85)** | **55.43 (10.48)** | **78.24 (6.67)** |
| | **Demographic Features** | | | 32.87 (13.81) | 53.68 (13.86) |
| **4m-PD** | RF | 71.00 (9.67) | 65.89 (11.96) | N/A | N/A |
| | TENER | 65.30 (23.25) | 58.57 (27.19) | N/A | N/A |
| | MDCSA$_1$ | *80.89 (9.16)* | *75.60 (12.79)* | **48.81 (13.52)** | *76.28 (14.09)* |
| | Ours RSSI | 77.47 (12.54) | 73.99 (13.00) | 41.79 (16.82) | 67.37 (16.86) |
| | DCMN | 74.43 (9.59) | 67.55 (14.50) | N/A | N/A |
| | Ours | **83.30 (6.73)** | **76.77 (13.19)** | *48.61 (12.03)* | **76.39 (12.23)** |
| | **Demographic Features** | | | 36.69 (18.15) | 50.53 (15.60) |

**Evaluation Metrics.** We are interested in developing a system to monitor PD progression in home environments. For example, we will consider if there is any significant difference in the performance of the systems when trained on a person with PD versus trained on someone without. Doing this may provide a useful insight whether there is any benefit to a person with Parkinson's to train our model with a training data from a healthy control (HC). We tailored our training procedure to test our hypothesis by performing variations of cross-validations. Apart from training our models on all HC subjects (ALL-HC), we also perform four different kinds of cross-validation: 1) We leave one PD subject out as training data (LOO-PD), 2) we leave one HC subject out as training data (LOO-HC), 3) We leave one HC subject and use only roughly four minutes worth of data as training (4m-HC), 4) We leave one PD subject and use only roughly four minutes worth of data as training (4m-PD). For all of our experiments, we test our trained models on all PD subjects (excluding the one used as training data for LOO-PD and 4m-PD). For room-level localisation accuracy, we use precision and weighted F1-score, all averaged and standard deviated across the test folds.

Table 2: Room-to-room transition accuracy of all models.

| Data | Models | Kitchen-Live | Kitchen-Dine | Dine-Live |
|---|---|---|---|---|
| **Ground Truth** | | 18.71 (18.52) | 14.65 (6.03) | 10.64 (11.99) |
| **ALL-HC** | RF | 16.18 (12.08) | **14.58 (10.22)** | 10.19 (9.46) |
| | TENER | 15.58 (8.75) | 16.30 (12.94) | 12.01 (13.01) |
| | MDCSA$_1$ | *16.83 (17.31)* | *14.91 (9.76)* | 9.81 (10.12) |
| | Ours RSSI | 15.73 (7.51) | 15.00 (10.64) | 11.82 (12.46) |
| | DCMN | 15.27 (7.51) | 13.40 (6.43) | *10.84 (10.81)* |
| | Ours | **17.70 (16.17)** | 14.94 (9.71) | **10.76 (9.59)** |
| **LOO-HC** | RF | 17.52 (16.97) | 11.93 (10.08) | 9.23 (13.69) |
| | TENER | 14.62 (16.37) | 9.58 (9.16) | 7.21 (10.61) |
| | MDCSA$_1$ | *17.68 (16.79)* | 14.14 (8.46) | 10.07 (13.55) |
| | Ours RSSI | 17.06 (15.34) | **14.38 (8.27)** | *11.04 (14.20)* |
| | DCMN | 16.30 (17.78) | 14.01 (8.08) | **10.37 (12.44)** |
| | Ours | **17.70 (17.42)** | *14.34 (9.48)* | 11.07 (13.60) |
| **LOO-PD** | RF | 14.49 (15.28) | 11.67 (11.68) | 8.65 (13.06) |
| | TENER | 13.42 (14.88) | 10.87 (10.37) | 6.95 (10.28) |
| | MDCSA$_1$ | 15.60 (14.02) | *14.16 (10.22)* | 8.89 (12.30) |
| | Ours RSSI | 15.60 (14.21) | 13.21 (9.65) | *10.07 (12.18)* |
| | DCMN | **16.98 (15.15)** | 15.26 (8.85) | 9.99 (13.03) |
| | Ours | *16.42 (14.04)* | **14.48 (9.81)** | **10.77 (14.18)** |
| **4m-HC** | RF | 14.22 (18.03) | 11.38 (15.46) | 13.43 (18.87) |
| | TENER | 10.75 (15.67) | 8.59 (14.39) | N/A |
| | MDCSA$_1$ | *19.38 (19.43)* | 16.34 (16.78) | 13.76 (17.47) |
| | Ours RSSI | 18.07 (16.78) | 16.72 (14.62) | 13.84 (18.36) |
| | DCMN | 16.89 (18.07) | **14.68 (13.57)** | *9.31 (15.70)* |
| | Ours | **18.15 (19.12)** | *15.32 (14.93)* | **11.89 (17.55)** |
| **4m-PD** | RF | 11.52 (16.07) | 8.73 (12.90) | N/A |
| | TENER | 8.75 (14.89) | N/A | N/A |
| | MDCSA$_1$ | 19.58 (18.62) | *14.30 (12.32)* | 13.47 (18.50) |
| | Ours RSSI | **19.00 (17.78)** | 16.12 (13.70) | *13.34 (17.69)* |
| | DCMN | 14.75 (13.79) | 13.47(17.66) | N/A |
| | Ours | *17.96 (19.17)* | **14.74 (10.83)** | **10.16 (14.03)** |

To showcase the importance of in-home mobility features in differentiating the medication state of a person with PD, we first compare how accurate the 'Room-to-room Transition' duration produced by each network to the ground truth. We hypothesis that the more accurate the transition to the ground truth the better mobility features are for medication state classification. For the medication state classification, we then compare two different groups of features for each network: 1) the baseline demographic features such as age, gender, years of PD, and PD assessment score specifically the Movement Disorders Society-sponsored revision of the Unified Parkinson's Disease Rating Scale motor subscore (MDS-UPDRS III), and 2) the normalised in-home mobility features. The metric we use for "ON" / "OFF" medication state evaluation is the weighted F1-Score and AUROC which are averaged and standard deviated across the test folds.

## 5.1 EXPERIMENTAL RESULTS

**Room-level Accuracy.** The first part of Table 1 compares the performance of our network and any other network for room-level classification. For the room-level classification, our network outperforms other networks and RF with a minimum improvement of 1.3% for the F1-score over the second best network (i.e. DCMN) in each cross-validation type with the exception of the ALL-HC validation. The improvement is more significant on the 4m-HC and 4m-PD validations, when the training data are limited, with an average improvement at almost 9% for the F1-score over the DCMN.

The LOO-HC and LOO-PD validation show that a model that has the ability to capture the temporal dynamics across time steps (e.g. TENER and DCMN) will perform better than a standard baseline technique such as Random Forest. DCMN performs better in those two validations due to its ability to capture asynchronous relation across modalities Jovan et al. (2022). However, when the training data become limited as in 4m-HC and 4m-PD validations, having extra capabilities is necessary to further extract temporal information and correlations. Due to being a vanilla transformer requiring considerable amount of training data, TENER performs worst in these two validations. DCMN performs quite well due to its ability to capture local context via LSTM for each modality and suppress a noisy modality, i.e. noisy accelerometer data via GRN. Our proposed network has all the capabilities that DCMN has with an improvement in the local context matching across modalities via DCSA. The local context matching capability seems to have a big impact in maintaining the performance of the network when the training data is limited. It is shown by how the $MDCSA_1$ outperforms the DCMN by an average of 7% for the F1-score in 4m-HC and 4m-PD validations.

**Room-to-room Transition and Medication Accuracy** The second part of the Table 1 shows the performance of all our networks for medication state classification. The **Demographic Features** can be used as a baseline for each type of validation. Our network, with the exception of the ALL-HC validation, outperforms any other network by a significant margin for the AUROC score. A minimum of 15% improvement over the baseline demographic features can be obtained by using the in-home mobility features produced by our model trained on ALL-HC training data. In 4m-PD validation data, demographic features show that its performance is not better than a simple random binary function with an AUROC score of 50.53% while our proposed network manages to achieve a 76.39% AUROC score with just 7.5% down from its best score in the LOO-HC validation. In the same validation, RF, TENER, and DCMN could not manage to provide any prediction due to their inability to capture (partly) hall transitions which are the main features for data samples for medication state classification. Furthermore, TENER model has shown its inability to provide any medication state prediction from the 4m-HC data validations. It can be traced back to Table 2 where the TENER model can not capture any transition between dining room and living room across all PD participants resulting incomplete features for all samples in 4m-HC validations. Our proposed network (and all its variations) are able to provide medication state prediction and maintain its performance across all cross-validations thanks to the addition of Equation 11 in the loss function calculation which forces the model to predict correctly the hall transitions. This is validated in Table 2 which shows that our proposed network, on average, has the closest transition duration prediction to the ground truth beating the second best by around 0.5 seconds across all hall transitions[3].

## 6    CONCLUSION

We have presented a new approach for indoor localisation utilising RSSI and accelerometer data. Our approach is based on (1) a novel transformer structure, which is carefully designed to take dual time-series inputs in combination with an CRF layer enforcing correct transition among time steps, and (2) an alteration to the loss function proposed in Jovan et al. (2022) to further strengthen the room-to-room transition prediction. We demonstrated that our proposed model is able to outperform the Dual Modality Context Network (DCMN) Jovan et al. (2022), and achieve SOTA results in room-level predictions on data collected of people with and without PD living freely in a smart home.

Within naturalistic settings, in-home mobility can be measured using our proposed approach, and we hypothesize that a person with PD during their "OFF" medication state shows signs of slowing in how they transition between rooms in their house. We show that, through our network which is a combination of multiple dual convolutional self-attention (MDCSA) and a CRF model, having an accurate indoor localisation system will give accurate in-home mobility features. This will lead to an accurate classifier model which can differentiate succinctly if a person with PD is in an "ON" or "OFF" medication state. We demonstrated that our proposed model is able to produce accurate in-home mobility features, which result in outperforming in-home mobility features produced by other models with an average improvement over 10% to the second best in AUROC score.

---

[3]The average ignores the 4m-PD validations since RF and DCMN does not have (some) hall transitions

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
