# OpenReview forum: "Indoor Localisation for Detecting Medication Use in Parkinson's Disease"
_ICLR.cc/2023/Conference — Submitted to ICLR 2023_

### Official Review · Reviewer_Ktqs · 2022-10-17

**Confidence:** 3
**Correctness:** 1
**Technical Novelty And Significance:** 1
**Empirical Novelty And Significance:** 1
**Recommendation:** 1

**Clarity, Quality, Novelty And Reproducibility:**

I had a difficult time understanding the motivation or what exactly they did different than previous results (Jovan 2022) other than adding a CNN layer and CRF evaluation criteria.

**Details Of Ethics Concerns:**

I don't see anything mentioned about IRB. If they are collecting a dataset I would need to see that there's IRB approval.

**Strength And Weaknesses:**

Strength:
New dataset with medication states (maybe?)

Weaknesses:
Poor writing
lack of results
difficult to understand motivation for methodical changes to their model

**Summary Of The Paper:**

The paper proposes a new model to evaluate Parkinson's disease feature extraction. In particular room-to-room classification. I'm not sure if the paper proposes a new dataset, but I think they do (it is not clear from the writings). In the experimental sections there's very close performances and very high standard deviations, it is unclear to me if the proposed methods works better. I have the following more detailed comments below:

"Indoor localisation, including its in-home mobility features" this sentence does not make sense to me.
"To improve the effectiveness of current methods for indoor localisa-
tion, a transformer-based approach utilising multiple modalities, Received Signal
Strength Indicator (RSSI) and accelerometer data from wearable devices, which
provide complementary views of movement, is proposed." please rephrase this is hard to read.
"To properly evaluate our proposed method, we use a free-living dataset where the movements and mobility are greatly varied and unstructured as expected in real-world conditions. " What's a free-living dataset? is this a standard to define such a dataset? please provide references in your commentary.
"12 pairs of people (one with PD, and the other a control participant) lived for five days
in a smart home with various sensors." Is this your own dataset? what is "various sensors"?
Reading the abstract it's clear that the authors are either novices, not native english speakers, or the PI didn't bother to help writing it. It's cluttered and cumbersome to digest. It should be clear, crisp, and spot-on.

"Although PD is a slowly progressive disease, the symptoms can fluctuate hourly depending on medication intake timing, stress and other factors." citation? where do you get this information from?

"These fluctuations make it challenging to capture the slow symptom progression in an individual (needed for example to measure how a new therapy changes disease progression) accurately over time using infrequent clinic or lab-based “snapshot” clinician-patient interactions. " this sentence is hard to read, what are you trying to say?

"Also, to help a clinician understand the impact of PD on a patient, continuous monitoring is needed to give an accurate evaluation of how severe the symptoms and their fluctuations are for that individual." While I can intuitively think that this might be reasonable for a practitioner to know this information, I want a research study motivating it. If not, why are you then doing this research project?

"Indoor localisation can be used to measure transitions between rooms in a home" can it? who says that? again, if you don't back up your statements with research papers I can't trust them. I don't know by default the state of the indoor localisation field. Moreover, the trailing citation is one 13 year old symposium paper on the "potential" of contextual awareness from wearables. I would expect a, somewhat recent, clinical journal paper formalizing the treatment procedure of Parkinson.

"To perform indoor localisation in home environments, IoT-based platforms with multimodal sensors combined with machine learning can be used to provide an unobtrusive and continuous localisation." I'm sure they can, but you need to prove it. Please provide a paper documenting this feature.

"Typically, a wearable inertial measurement unit (IMU) is used to produce a received signal strength indicator (RSSI) at points throughout a home, which can be used to create radio-map features for room-level localisation." If it's a typical setup, then you should have plenty of studies, or protocols, detailing it. Provide these.

At the end of your introduction, it seems like you work off the Jovan 2022 paper. It's not clear to me how you differ from their research findings. Make your motivation more clear in the introduction for what motivated your to alter their method, what you changed, why you believed those changes are important, and how much better you perform.

"Early work in machine ..." you mention "simple PD classification". I'm not a PD researcher, I don't know what that means. This sentence does not make it clear to me how the initial models on PD prediction operates.
For the vision models you mention, what are they predicting? Why were they better? Is the main concern privacy?

"RSSI signals are not stable, they fluctuate randomly due to shadowing, fading and multi-path effects." I need a citation here.

What wearable are you using in your study, is it a fitbit? and apple watch?

"To measure the accuracy ML models" I don't understand this

What did you use the camera data for? did a person manually sit and look it through to annotate the gold label for when participants when to different rooms? how did this work at night?

Are you preprocessing steps standard in the field? why don't you cite any papers?

Your data preprocessing is not completely clear to me.

For your dataset, did you collect it yourself, or where is it from? if you collected it yourself, where's your IRB? Why did you need to gather a new dataset, was it just for medicatio ON/OFF?

The math explaining your model is confusing, it doesn't give me a clear overview of how your model differs from the Jovan 2022 paper. As far as I understand your modification is applying a CNN layer at the start and a CRF layer at the end?

Table 2, ALL-HC ours the performance reported is 17.70 (16.17). Does that mean you have 17.70 mean accuracy and 16.17 standard deviation?

How many of your results are statistically significant?

**Summary Of The Review:**

At this stage, I do not believe this article is suitable for publication.  Moreover, their results do not seem significant and their standard deviations are very large.

---

### Official Review · Reviewer_NCB2 · 2022-10-24

**Confidence:** 4
**Correctness:** 2
**Technical Novelty And Significance:** 3
**Empirical Novelty And Significance:** 3
**Recommendation:** 3

**Clarity, Quality, Novelty And Reproducibility:**

The paper contains some technical errors. In the beginning it is stated that IMU provides RSSI measurements. IMU (Inertial Measurement Unit) contains 3-axis accelerometer and 3-axis gyroscope, therefore it provides measurements of specific force and angular rate. The accelerometer data is said to have dimensions of 5x6, where 5 is the measurement rate (5Hz). I guess you are actually inputting the full IMU measurement and not just accelerometer as is stated?  interpretation of the results is not very clear.

**Details Of Ethics Concerns:**

The paper discusses tracking of people at their homes and therefore the handling of collected data should be discussed.

**Strength And Weaknesses:**

The paper addresses relevant topic, and the developed method seems interesting and technically sound.  However, the experiment setup and results raise many questions. Is it enough to see how the medication affects the patient’s motion in five days?  Transfer between some rooms is inferred correctly, whereas for some not. The interpretation of the results is not very clear. The results show that the method outperforms a benchmark that has been considered as the state-of-the-art method. However, some more traditional methods outperform the developed one. Therefore, the effect seems to be quite incremental.

**Summary Of The Paper:**

This paper discusses a method for detecting the evaluation Parkinson’s disease and the patients’ use of medication by monitoring the motion of patients at home. This is done by indoor localization at room level fusing Bluetooth RSSI measurements and accelerometer (or actually whole IMU?) measurements. The paper builds on state-of-the-art method based on Dual Context Modality Network by replacing its LSTM network with causal convolutional layers and self-attention (DCSA).

**Summary Of The Review:**

The paper is interesting, but it includes technical errors. Although the developed method is relevant, the experiments and its analysis leave more questions than give answers.

---

### Official Review · Reviewer_r87h · 2022-10-24

**Confidence:** 3
**Correctness:** 3
**Technical Novelty And Significance:** 2
**Empirical Novelty And Significance:** 2
**Recommendation:** 3

**Clarity, Quality, Novelty And Reproducibility:**

The novelty of the paper is very limited
- The paper only marginally modifies the previous model architecture for indoor localization of Jovan et al. 2022.
- The additional classification step uses feature extracted from the localization passing them onto a standard classifier - this step may be useful for the application area but is not anyhow novel.

Reproducibility is very limited
- neither the dataset nor the code has been released (and the authors do not indicate their availability anyhow)

Quality of experimental evaluation could be improved
- Why is accuracy of medication state not reported?
- Standard deviations seem rather high, most reported improvements seem statistically insignificant. Any comments?
- What are teh demographic features and how are they used to predict the medication state?

**Details Of Ethics Concerns:**

The experimental setup collect data from Parkinson disease patients and control subjects living in a smart house. The Parkinson disease patients are moreover asked to go off mediation in order to be able to collect relevant data. I presume the consent of the patients have been thought before but the paper does not clarify this (or I could not find it). Not sure if any consent of a relevant ethical commission is required for this type of experimentation and data collection.


**Strength And Weaknesses:**

Strength:
The paper has a strong motivation of using indoor localization data to improve non-intrusive health monitoring of PD patients.
It uses data from a well designed experimental setup with real patients and control subjects.
It seems to perform slightly better than previous methods

Weaknesses:
The contribution of the paper is rather little - it largely bases on previous paper of Jovan at all 2022, reusing their motivation, problem definition as well as data.
The method is only slightly modified reaching small, insignificant performance improvement.
The paper is not reproducible (see next section)

**Summary Of The Paper:**

The paper proposes a method to improve indoor localization based on combination of RSSI and accelerometer data from waerable device. The localization data is used to produce features that serve as inputs for a classification model aiming to recognize when patients with Parkinson disease are off their medication. The analysis is performed or real data collected from PD patients and healthy control subjects living in a smart home. The localization model is based on the previous work of Jovan et al. 2022 changing their NN architecture from RNN with attention to causal convolution with attention. Furthermore, they complement the outputs at the room-identification level and room-transition counts by the actual on/off medication classification, reporting favorable results.

**Summary Of The Review:**

The paper addresses an interesting and important application problem. However, its contribution is marginal mainly due to being largely based on on work of Jovan et al. 2022 bringing little new. Neither the dataset nor the method have been released and therefore cannot serve the community for further investigation. I can imagine the paper being of interest to the specific application community focusing on PD patients. In my view, however, it does not have the potential to trigger interest from the ICLR community and does not contribute enough to be considered for publication. I therefore recommend to reject the paper.

---

### Official Review · Reviewer_Q2sw · 2022-10-27

**Confidence:** 4
**Correctness:** 3
**Technical Novelty And Significance:** 2
**Empirical Novelty And Significance:** 3
**Recommendation:** 3

**Clarity, Quality, Novelty And Reproducibility:**

The paper is of high quality and clarity but its originality could be disputed. There are multiple works in the biomedical processing community focused on similar topics.

**Strength And Weaknesses:**

S

+ important ML application
+ solid analysis of the dataset (albeit small for modern ML)


W

+ limited model novelty (reusing existing parts)
+ limited new insights for the representation learning community

**Summary Of The Paper:**

This work proposes an approach for indoor localisation utilising RSSI and accelerometer data based on transformers with CRF layer an alternating loss function. Its contributions are mostly in the specific application (indoor localisation) rather than the methods.

**Summary Of The Review:**

I believe this paper would be a better match in a biomedical/mobile computing venue.

---

### Decision · Program_Chairs · 2023-01-20

**Decision:**

Reject

**Justification For Why Not Higher Score:**

Not good enough

**Justification For Why Not Lower Score:**

N/A

**Metareview: Summary, Strengths And Weaknesses:**

All reviewers agree that the proposed method is not original, lacks clarity and reproducibility and may not be a good fit for ICLR.